# Niche-Driven Bacterial Assembly Versus Weak Geographical Divergence of Fungi in the Rhizosheath of Desert Plant *Leymus racemosus* (Lam.) Tzvel

**DOI:** 10.3390/plants14243747

**Published:** 2025-12-09

**Authors:** Yufang Sun, Jinfeng Tang, Xiaohao Zhou, Jun Liu

**Affiliations:** 1Xinjiang Key Laboratory of Soil and Plant Ecological Processes, Xinjiang Agricultural University, Urumqi 830052, China; 13579835054@163.com; 2College of Life Sciences, Xinjiang Agricultural University, Urumqi 830052, China; c1330492964@163.com (X.Z.); liujem@126.com (J.L.)

**Keywords:** *L. racemosus*, rhizosheath, microbial community, habitat, geographical distance

## Abstract

The rhizosheath plays a critical but poorly understood role in plant–microbe interactions. However, it still remains unclear how host selection versus geographical isolation contributes to microbial community assembly within the rhizosheath. This study characterized the bacterial and fungal communities in the rhizosheath and surrounding bulk soil of *Leymus racemosus* using 16S rRNA and ITS high-throughput sequencing. Results showed that the bacterial community was strongly shaped by host selection within the rhizosheath, based on significantly reduced α-diversity and distinct β-diversity (Permutation tests, *p* < 0.001) compared to bulk soil. Furthermore, the core bacterial community structure was highly similar between the two geographically separated sites (PERMANOVA, *p* = 0.089). In contrast, the fungal community exhibited weaker habitat specificity but showed significant, though weak, geographical divergence (β-diversity, Permutation tests, *p* = 0.028). The explanatory power of geographical distance for fungal community variation was low (R^2^ = 0.095) and less than that of the rhizosheath microhabitat (R^2^ = 0.142). In conclusion, the rhizosheath imposes a strong filtering effect on bacterial communities. The weaker habitat specificity and stronger geographical signal observed for fungi suggest potential regulation by local dispersal limitation or historical colonization processes. This study provides insights into the assembly mechanisms of the plant rhizosphere microbial community.

## 1. Introduction

The rhizosheath is a highly specialized adaptive structure that has evolved in gramineous plants as an adaptation to arid and nutrient-deficient environments [1,2]. It is formed through the synergistic binding of root exudates, soil particles, and microorganisms, creating a multifunctional micro-ecosystem [2,3]. Studies indicate that the rhizosheath plays several critical roles in mitigating extreme edaphic stressors: its physical architecture not only extends the root–soil interface—thereby significantly improving plant capacity for water and nutrient uptake—but also offers mechanical protection to roots and enhances plant anchorage in erosion-prone settings. Moreover, the rhizosheath serves as a selective microbial habitat. Its distinct microenvironmental properties support the colonization and assembly of specific microbial consortia, playing a central role in regulating plant–microbe interactions [4,5,6].

The assembly mechanisms of rhizosheath microbial communities constitute a central theme in microbial ecology, yet key scientific questions regarding these processes remain unresolved. A particularly critical issue involves identifying the dominant forces that govern community assembly across varying spatial scales, which is fundamental to deciphering the principles underlying microbial community organization [7,8]. These assembly processes are generally regulated by a combination of ecological forces—primarily environmental selection, dispersal limitation, and ecological drift [9,10]. Studies indicate that the relative importance of these driving factors shifts dynamically with changes in spatial scale [11]. At large spatial scales, such as continental or regional gradients, dispersal limitations imposed by geographical isolation and environmental heterogeneity—including variations in soil type and climate—often emerge as the dominant structuring factor [12]. Under such conditions, strong environmental filtering tends to override the more subtle influences of plant-specific rhizosphere effects.

As the spatial scale diminishes, the influence of the rhizosphere effect becomes increasingly prominent. At smaller spatial scales, such as within a single field under a homogenized edaphic background, the host plant exerts a dominant selective pressure through its genotype, root architecture, and exudate profile. This pressure serves as a key driver structuring the specific rhizosphere microbiota, ultimately leading to significant community divergence from the bulk soil [13].

At the mesoscale—spanning distances from tens to hundreds of kilometers within ecologically coherent units such as a watershed or biome—the mechanisms governing microbial assembly often become increasingly complex and remain poorly resolved [14,15]. At this intermediate spatial extent, both dispersal-related processes (shaped by geographic distance and the resulting heterogeneity in regional species pools) and contemporary selection by plant roots may exert comparable, interactive influences on community composition [16]. On one hand, phylogenetically conserved host plants can recruit a conserved core microbiota across geographically discrete but edaphically similar sites [17]. On the other hand, divergence in the initial regional species pool may lead to pronounced compositional differences in the assembled rhizosphere community [18,19]. Existing evidence suggests that geographical distance generally exerts a stronger influence on fungal community variation than on bacterial communities, primarily due to stronger dispersal limitations and specialized interactions with host plants in fungi. In contrast, bacterial communities are more strongly governed by local environmental filtering [20]. However, at the mesoscale, the relative contributions of plant selective regulation and geographical isolation to the assembly of different microbial groups, such as bacteria and fungi, remain insufficiently explored.

Based on the above background, this study selects the typical desert plant *Leymus racemosus* (Lam.) Tzvel as the research subject. This species has evolved a well-developed rhizosheath structure during its long-term adaptation to extreme environments (Figure 1). Previous evidence indicates that the rhizosheath of *L. racemosus* plays a key role in enhancing drought tolerance and maintaining ecosystem function under water stress [21]. rendering it an ideal model system for investigating plant–microbe interactions. The distribution of *L. racemosus* in China is relatively limited, found mainly in the Junggar Basin and the Irtysh River basin in Xinjiang (Figure 1a,b). This study established two sampling sites approximately 17 km apart within the Kalamaili Nature Reserve in Xinjiang. The choice of this distance is scientifically justified: it is sufficient to detect geographical dispersal effects at the mesoscale, while simultaneously ensuring high consistency between the two sites in terms of large-scale environmental factors such as macro-climate and soil type. This allows for the effective isolation of the influences of geographical distance and the rhizosheath microenvironment from complex environmental heterogeneity, focusing specifically on the relative importance of plant selection versus spatial isolation in shaping the rhizosheath microbial community under homogeneous environmental background conditions.

Based on this research background, we propose the following hypothesis: In a homogeneous habitat at the mesoscale, the rhizosheath microenvironment of *L. racemosus* may exert a strong selective effect on microbial community assembly, but different microbial groups (e.g., bacteria vs. fungi) may exhibit divergent response patterns to geographical distance and the rhizosheath microenvironment. By validating this hypothesis, this study provides insights into the mechanisms governing plant rhizosphere microbiome assembly.

## 2. Results

### 2.1. Soil Properties

Analysis of the soil’s physical and chemical properties revealed that the two habitats were highly comparable. The pH, electrical conductivity (EC), and concentrations of total organic carbon (TOC), total nitrogen (TN), and total phosphorus (TP) showed no statistically significant differences (Table 1). Furthermore, the concentrations of major ions (K^+^, Na^+^, Ca^2+^, SO_4_^2−^, and Cl^−^) were also consistent between habitats, with no significant differences detected (one-way ANOVA, *p* > 0.05).

### 2.2. Microbial Community Overview

High-throughput sequencing of the 16S rRNA gene yielded 2,161,220 raw reads. After quality filtering, 2,020,081 high-quality reads were retained, which were clustered into 10,262 operational taxonomic units (OTUs) at a 97% similarity threshold. For the fungal ITS region, 1,563,786 raw reads were obtained, yielding 1,478,939 clean reads that were clustered into 949 OTUs.

Sankey diagram analysis revealed that the rhizosheath bacterial community composition remained highly consistent between the two habitats, with no significant differences in the relative abundance of dominant taxa (Figure 2a). The bacterial community was predominantly composed of *Actinobacteria*, *Alphaproteobacteria*, *Bacilli*, and *Bacteroidia*. In contrast, the fungal community exhibited considerable spatial variation (Figure 2b). Across the geographical gradient, significant structural divergence was observed: rhizosheath samples from Habitat 1 (R1) were mainly enriched in *Agaricomycetes* and *Dothideomycetes*, whereas those from Habitat 2 (R2) showed significant enrichment in *Dothideomycetes*, *Eurotiomycetes*, *Sordariomycetes*, and *Tremellomycetes*.

### 2.3. Characteristics of the Bacterial Community

#### 2.3.1. Bacterial Community Composition

Analysis of bacterial α-diversity revealed no significant difference in the Shannon index between the rhizosheath samples from the two habitats (Kruskal–Wallis tests, *p* > 0.05), indicating comparable community diversity. However, the rhizosheath exhibited a significantly lower Shannon index compared to the bulk soil (Kruskal–Wallis tests, *p <* 0.05), suggesting a strong reduction in bacterial diversity due to the plant selection effect (Figure 3a,b). Principal coordinate analysis (PCoA) based on bacterial communities showed that rhizosheath samples from both habitats formed a tight cluster, demonstrating high compositional similarity (Permutation tests, *p* = 0.161) (Figure 3c). In contrast, clear separation was observed between rhizosheath and bulk soil samples (Permutation tests, *p* = 0.001) (Figure 3d). These results were corroborated by PERMANOVA, which confirmed no significant difference between the two rhizosheath communities (Permutation tests, R = 0.256, *p* = 0.089) but a highly significant difference between the rhizosheath and bulk soil communities (Permutation tests, R = 0.9, *p* = 0.0002) (Figure 4a,b).

#### 2.3.2. Core Rhizosheath-Specific Bacterial Taxa

Analysis of the core bacterial communities in the *L. racemosus* rhizosheath revealed that, although persistent core taxa represented only approximately 28% of the total OTUs, they nevertheless dominated the community, comprising over 80% of the total relative abundance (Figure 5). At the phylum level, the rhizosheath was predominantly composed of *Actinobacteriota*, *Proteobacteria*, *Bacteroidota*, and *Firmicutes*. At the genus level, the dominant taxa included *Arthrobacter* (20.35%), *Paenarthrobacter* (3.97%), and *Pedobacter* (3.78%). In contrast, the bulk soil communities were primarily characterized by *Actinobacteriota*, *Acidobacteriota*, and *Chloroflexi* at the phylum level and by *Rubrobacter*, unclassified norank_o__0319-7L14, and *Arthrobacter* at the genus level (Figure 6a,b).

Functional prediction of the core microbial community in the rhizosheath, conducted using FAPROTAX, revealed a significant enrichment of chemoheterotrophic microorganisms. These microbes utilize organic compounds released by *L. racemosus* roots as both a carbon and energy source, playing a crucial role in rhizosphere carbon cycling and energy flow. Concurrently, the rhizosheath was highly enriched with functional groups associated with nitrate reduction and urea decomposition. These microorganisms convert nitrogen forms that are less accessible to plants, such as nitrate and urea, into preferred nitrogen sources such as ammonium nitrogen. This process enhances nitrogen availability in the rhizosphere, thereby providing critical nutritional support for plant growth and development (Figure 7).

### 2.4. Fungal Community Characteristics

#### 2.4.1. Fungal Community Composition

Fungal α-diversity, as measured via the Shannon and Chao indices, showed no significant differences in the rhizosheath between the two habitats (Kruskal–Wallis tests, *p* > 0.05) (Figure 8a,b). Principal coordinate analysis (PCoA) further demonstrated significant compositional divergence of the fungal communities between habitats (Permutation tests, *p* = 0.028; Figure 8c). This finding was supported by PERMANOVA, which confirmed a significant difference in fungal community structure between the two rhizosheath habitats (Permutation tests, R = 0.396, *p* = 0.0249; Figure 9a). Moreover, highly significant differences were observed between the rhizosheath and bulk soil fungal communities (Permutation tests, R = 0.355, *p* = 0.002; Figure 8d) and were supported by PERMANOVA (Permutation tests, R = 0.455, *p* = 0.001; Figure 9b).

#### 2.4.2. Analysis of Differences in Fungal Communities

Analysis at the genus level revealed distinct fungal community compositions between the two habitats. Both community composition and differential abundance analyses indicated that the rhizosheath of *L. racemosus* at Habitat 1 (R1) was predominantly enriched with *Curvularia* (26.4%), *Moniliophthora* (16.5%), and *Stropharia* (11.6%). In contrast, the rhizosheath at Habitat 2 (R2) was primarily characterized by the enrichment of *Talaromyces* (17.5%), *Preussia* (14.3%), and *Chaetomium* (9.6%) (Figure 10).

**Figure 8 plants-14-03747-f008:**
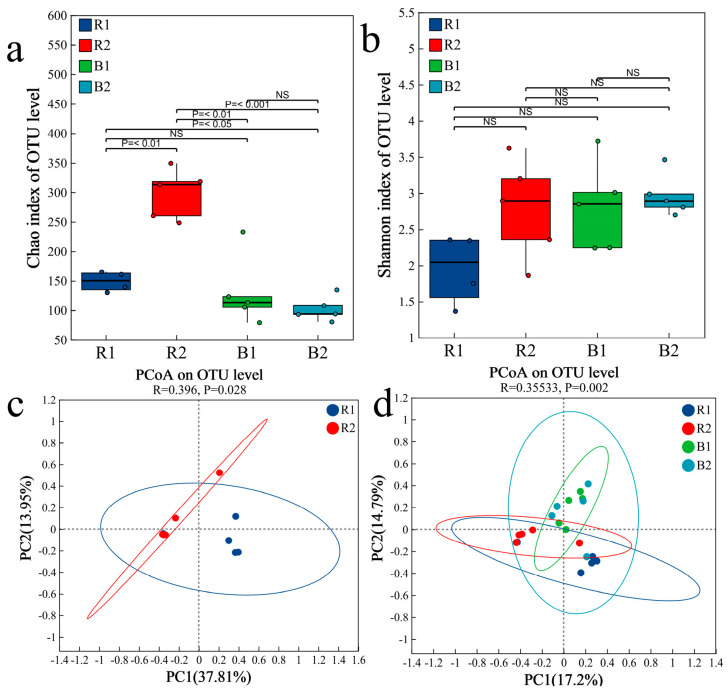
The values of the α−diversity indices for fungal communities (**a**,**b**) are shown as boxplots for the two habitats. R1 and R2 represent rhizosheath samples collected from Habitat 1 and Habitat 2, respectively; B1 and B2 refer to bulk soil samples obtained from the corresponding habitats; *p* ≤ 0.05, significant; NS, not significant. PCoA showing the fungal community structure based on distinct samples of the two habitats (**c**) and based on rhizosheath and bulk (**d**).

**Figure 9 plants-14-03747-f009:**
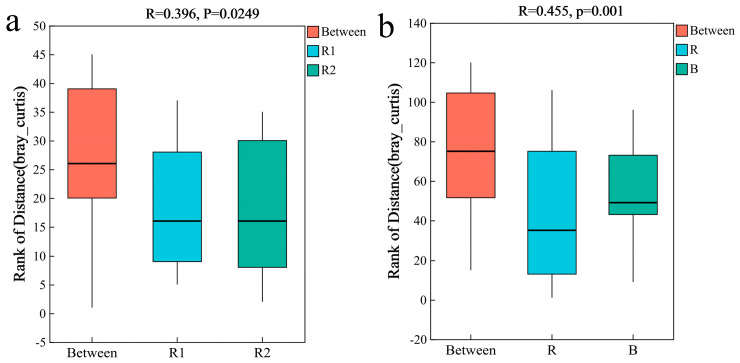
PERMANOVA among different groups. (**a**) Differences in rhizosheath fungi between the two habitats; (**b**) Differences between rhizosheath and bulk soil fungi. The R-value is closer to 1, indicating that the difference between groups is more significant than the difference within groups; the smaller R-value indicates no significant difference between and within groups. R: rhizosheath soil samples; B: bulk soil samples.

**Figure 10 plants-14-03747-f010:**
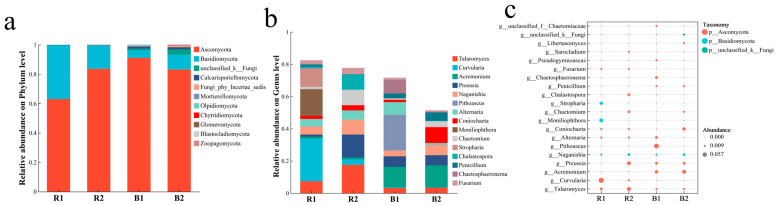
(**a**,**b**): Composition of the predominant fungi in the rhizosheath and bulk soil across the two habitats at the phylum and genus levels. (**c**): Bubble plot of differentially abundant fungal taxa, with bubble size and color intensity representing taxonomic identity and abundance levels across samples.

### 2.5. Decomposition of Multifactorial Explanatory Power

To quantify the effects of the rhizosheath microenvironment and geographical distance on microbial community variation, a PERMANOVA based on OTU-level data was performed (Table 2). The results, expressed as R^2^ values, represent the proportion of community variance explained by each factor, with statistical significance set at *p* < 0.05.

The analysis showed that the rhizosheath microenvironment independently explained 46.6% of the variance in bacterial community composition, which was substantially greater than that explained by geographical distance alone (5.54%). The variance shared between the two factors was 5.86%. In contrast, for the fungal community, the rhizosheath microenvironment explained a lower proportion of the variance (14.2%). Meanwhile, the independent contribution of geographical distance increased to 9.51%, with the shared variance accounting for 7.19% (Figure 11).

## 3. Discussion

### 3.1. The Strong Microenvironment Selection in the Rhizosheath Bacterial Community of L. racemosus

In desert ecosystems characterized by harsh and extreme conditions, plants often enhance stress resilience by establishing symbiotic relationships with rhizosphere microorganisms [22,23,24]. The structure and diversity of these microbial communities are influenced by a complex array of factors [25,26]. At broad spatial scales, geographic distance drives heterogeneity in climatic and edaphic conditions—such as soil type and elevation—intensifying environmental filtering and establishing it as the dominant force shaping microbial assembly [27]. In contrast, under localized conditions with minimal environmental variation, the “microenvironment selection hypothesis” posits that host-mediated filtering becomes the central mechanism of community assembly [28,29,30]. Through root exudates and associated signaling pathways [28], plants selectively recruit beneficial microorganisms and create specialized ecological niches, thereby partially overcoming the limitations on microbial dispersal imposed by geographic isolation [31].

This study reveals that in environmentally homogeneous habitats—where large-scale factors such as altitude and climate are consistent, and no significant differences in soil physicochemical properties are observed (Table 1)—the composition and diversity of the bacterial community in the rhizosheath of *L. racemosus* remain highly consistent despite a geographical separation of 17 km. Further analysis indicated that the rhizosheath microenvironment explained 46.6% of the bacterial community variation, a contribution substantially greater than that of geographical distance. These results strongly support the microenvironment selection hypothesis, suggesting that within a certain spatial scale, when large-scale environmental filtering is uniform, geographic isolation exerts only a limited influence on the assembly of the core bacterial community in the rhizosheath of *L. racemosus*. This finding aligns with those reported by Liu et al. [32]. Moreover, significant differences in microbial community structure were detected between the rhizosheath and bulk soil, further confirming that rhizosheath bacterial assembly is primarily driven by microenvironmental conditions and underscoring the dominant role of the host plant in structuring the rhizosheath microbiome. In contrast, a study by Xu et al. reported and demonstrated that geographical distance significantly reduced the similarity of rhizosphere bacterial communities in rubber trees [33]. Although this conclusion diverges from the present results, it highlights the scale-dependent nature of geographic effects: while its influence may be limited within restricted spatial ranges, beyond a certain threshold, geographic isolation can become a dominant factor driving microbial divergence [11,12].

As a dominant species in the Karamaili Desert, *L. racemosus* secretes root mucilage that adheres to soil particles on the root surface, forming a typical rhizosheath structure. This structure not only acts as a physical barrier that restricts the free diffusion of microorganisms [2] but also helps maintain internal humidity while limiting the random invasion of external bacteria. These mechanisms may mitigate the dispersal limitations imposed by geographic distance, thereby promoting the stability of the bacterial community within the rhizosheath [34].

Chemical selection within the rhizosheath is also considered a key mechanism shaping the specific microbial assemblage associated with *L. racemosus* [35]. Previous studies have shown that plant roots can actively modulate the recruitment of beneficial microorganisms through the secretion of specific compounds [35,36]. As a dominant desert species, *L. racemosus* likely facilitates the targeted enrichment of functional taxa such as *Actinobacteriota* and *Proteobacteria* via root exudates. This metabolically driven selection mechanism may enable the host plant to preferentially recruit microorganisms with drought tolerance and plant growth-promoting traits, thereby enhancing its adaptability to extreme environments [37,38].

This study also identified significant enrichment of bacterial genera such as *Arthrobacter*, *Paenarthrobacter*, and *Pedobacter* in the rhizosheath of *L. racemosus*. Among these, *Arthrobacter* is widely recognized as a predominant genus in the rhizosphere of desert plants, exhibiting broad carbon utilization capacity and strong environmental adaptability. Numerous studies have confirmed that this genus possesses plant growth-promoting functions, including phosphate solubilization, nitrogen fixation, and potassium release, as well as ACC deaminase activity, antibiotic production, and siderophore synthesis. These traits play a vital role in enhancing stress resistance and facilitating nutrient acquisition in plants under desert conditions [39,40,41,42].

In summary, the assembly of the bacterial community in the rhizosheath of *L. racemosus* adheres to the microenvironment selection hypothesis. Within a moderate spatial scale, the influence of geographical distance on bacterial community differentiation can be offset by plant-mediated microenvironmental regulation. The synergistic effects of the physical barrier formed by the rhizosheath and the chemical compounds secreted by the roots are likely the primary mechanisms underlying the specificity and stability of its core bacterial community.

### 3.2. The Weak Influence of Geographical Distance on Rhizosheath Fungal Communities

In contrast to the bacterial community, the fungal community within the rhizosheath of *L. racemosus* exhibited discernible, though modest, variation with geographical location. Although statistical analysis indicated significant differences in fungal composition and diversity between the two sites (*p* = 0.028), PERMANOVA effect size estimation revealed that geographical distance explained only 9.51% of the community variation (R^2^ = 0.095). While this value is notably higher than its explanatory power for bacteria, it still reflects only a limited absolute influence of geographical isolation on fungal assembly. In comparison, the rhizosheath microenvironment accounted for 14.2% of the variation (R^2^ = 0.142), significantly lower than its dominant role in structuring bacterial communities. The combined explanatory power of these two factors reached only 30.9%, indicating that nearly 70% of the variation in fungal community composition is governed by other, unmeasured variables.

This pattern may be attributed to the distinct biological and ecological traits of fungi. First, limited spore dispersal is likely a key factor [43,44]. The 17 km separation between the two sites may exceed the effective dispersal range of many fungal spores in arid environments, thereby restricting gene flow and promoting community differentiation [45,46]. Nevertheless, the observed degree of differentiation remained relatively low. We speculate that the presence of widely distributed taxa and/or highly efficient dispersal mechanisms may partially counteract the effects of geographical isolation, although this hypothesis warrants further investigation. Second, fungal community assembly may be influenced by historical colonization and succession processes [47,48]. The two sites may have experienced distinct initial colonization events or divergent successional trajectories during early community formation. Even under similar contemporary environmental conditions, such historical contingencies could lead to persistent structural differences. These priority effects may sustain detectable—though biologically limited—spatial differentiation even in the absence of major environmental variation.

Analysis of the rhizosheath fungal community in *L. racemosus* across geographic sites revealed consistent enrichment of key genera such as *Talaromyces*, *Curvularia*, *Naganishia*, *Stropharia*, and *Chaetomium*, although their relative abundances varied spatially. These taxa constitute a functionally diverse reservoir of microorganisms with potential biotechnological relevance. Specifically, *Talaromyces* and *Curvularia* act as effective phosphate solubilizers and soil protectors. They not only enhance the availability of insoluble phosphorus—reducing reliance on chemical fertilizers—but also secrete antimicrobial compounds that suppress soil-borne pathogens such as *Verticillium* [49,50,51,52]. *Naganishia*, recognized for its stress tolerance, exhibits extreme environmental adaptability, providing valuable germplasm resources for developing microbial inoculants that enhance crop resilience to drought and salinity [53]. Meanwhile, *Stropharia* demonstrates a strong capacity to degrade recalcitrant organic compounds, highlighting its potential in agricultural waste recycling and bioremediation of organically polluted soils [54]. *Chaetomium*, well-established as a biocontrol and plant growth-promoting agent, has relatively mature commercialization pathways, offering a practical route for direct application of rhizosheath-derived beneficial fungi [55,56].

In summary, the assembly of the fungal community in the rhizosheath of *L. racemosus* appears to be governed by mechanisms distinct from those influencing bacteria, involving a more complex and multifactorial process. While both microenvironmental filtering and limitations on geographic dispersal exert discernible yet relatively weak influences, fungal community structure is likely shaped predominantly by other biological or historical factors not captured in this study.

### 3.3. Study Limitations

While this study reveals the distinct assembly mechanisms of bacterial and fungal communities in the rhizosheath of *L*. *racemosus* within a homogeneous habitat, several limitations that constrain the interpretation of our findings should be considered. Firstly, the research design primarily focused on deterministic processes (such as environmental filtering and dispersal limitation) and did not fully evaluate the influence of stochastic processes (e.g., ecological drift) on community assembly. This is particularly relevant for the fungal communities, where nearly 70% of the variation remained unexplained, strongly suggesting that stochastic processes and priority effects may play a pivotal role. The communities at the two sites might have diverged due to minor differences in their initial colonization history, yet such historical contingencies could not be captured by our static sampling design. Secondly, the sample size and spatiotemporal scale were limited. This study involved only a single sampling event from two sites, failing to encompass seasonal dynamics or inter-annual fluctuations. Finally, our study lacks functional validation. Research based on 16S and ITS amplicon sequencing can only infer taxonomic composition and potential functions but cannot reveal the actual metabolic activities of key taxa (such as the enriched *Arthrobacter* or *Talaromyces*) or their specific interactions with the host plant. Subsequent studies integrating metagenomics, metatranscriptomics, and pure culture verification will be essential to elucidate how these microbes concretely contribute to the stress resistance physiology of *L. racemosus*, thereby transforming correlative inferences into causal mechanistic insights.

## 4. Materials and Methods

### 4.1. Study Area and Sampling

This study investigated two disjunct populations of *L. racemosus* in the Kalamaili Nature Reserve (Figure 1a). The two study sites, H1 (altitude: 983 m, 89.082322° E, 45.246486° N) and H2 (altitude: 998 m, 89.254483° E, 45.338486° N), represent typical desert habitats for this species and are separated by approximately 17 km. Site selection was based on their shared macroenvironmental conditions (e.g., climate) and comparable altitude, which establishes a baseline of similar abiotic factors. Subsequent soil analysis confirmed the absence of significant differences in fundamental physicochemical properties between the sites (Table 1). This configuration—featuring spatially isolated populations under comparable environmental regimes—provides a robust comparative system to disentangle the relative effects of geographical distance versus host selection in shaping the root-associated microbiome.

Soil sampling was performed in early May 2023. In each habitat, five 5 × 5 m quadrats were randomly established, maintaining a minimum distance of 20 m between quadrats to ensure spatial independence. From each quadrat, 5–8 healthy, similarly sized *L. racemosus* individuals were randomly selected. Soil profiles were excavated to a uniform depth (30–50 cm), corresponding to the root zone of the plants. The intact rhizosheath was carefully collected by shaking off loosely adhered soil and immediately placed into sterile sealed bags on ice, following established methods [57,58]. Concurrently, bulk soil was collected from the same root depth in each quadrat using a five-point sampling method.

All samples were transported to the laboratory on ice. Upon arrival, the rhizosheath soil was separated from the roots. For subsequent analysis, a composite rhizosheath sample per quadrat was created by combining equal amounts of soil from 4 to 5 plants within the same quadrat. Similarly, a composite bulk soil sample per quadrat was generated by mixing soil from the five sampling points. Each composite sample (both rhizosheath and bulk soil) was then divided into two aliquots: one was flash-frozen in liquid nitrogen and stored at –80 °C for DNA extraction and high-throughput sequencing, while the other was air-dried for soil physicochemical property analysis.

### 4.2. Analysis of Soil Physicochemical Properties

Soil pH and electrical conductivity (EC) were measured using a pH meter (FE20, Mettler-Toledo Instruments, Shanghai, China) and a conductivity meter (Hanna HI98192, Hanna Instruments, Cluj-Napoca, Romania), respectively. The concentrations of total organic carbon (TOC), total nitrogen (TN), and total phosphorus (TP) were determined following established protocols [59,60]. The levels of major ions (Na^+^, K^+^, Ca^2+^, Cl^−^, SO_4_^2−^) were quantified using an ion chromatograph (ICS-5000, Thermo Fisher Scientific, Waltham, MA, USA) coupled with inductively coupled plasma optical emission spectrometry (5110 ICP-OES, Agilent, Santa Clara, CA, USA).

### 4.3. DNA Extraction and High-Throughput Sequencing

Total microbial genomic DNA was extracted from bulk soil and rhizosheath soil samples (0.5 g each) using the E.Z.N.A.^®^ Soil DNA Kit (Omega Bio-tek, Norcross, GA, USA) in accordance with the manufacturer’s instructions. For each sample, three technical replicates were performed to ensure extraction consistency. The quality and concentration of the extracted DNA were assessed by 1.0% agarose gel electrophoresis and a NanoDrop 2000 spectrophotometer (Thermo Fisher Scientific, Wilmington, DE, USA), after which the DNA was stored at –80 °C until further processing.

For bacterial community analysis, the hypervariable V3–V4 region of the 16S rRNA gene was amplified with the primers 338F (5′-ACTCCTACGGGAGGCAGCAG-3′) and 806R (5′-GGACTACHVGGGTWTCTAAT-3′) [61]. For fungal community analysis, the ITS1 region was targeted using primers ITS1F (5′-CTTGGTCATTTAGAGGAAGTAA-3′) and ITS2 (5′-GCTGCGTTCTTCATCGATGC-3′) [62]. The PCR mixture (20 µL total volume) contained 4 µL of 5× Fast Pfu buffer, 2 µL of 2.5 mM dNTPs, 0.8 µL of each primer (5 µM), 0.4 µL of Fast Pfu polymerase, and 10 ng of template DNA, with ddH_2_O added to volume. Amplification was conducted under the following conditions: initial denaturation at 95 °C for 3 min, 27 cycles of denaturation at 95 °C for 30 s, annealing at 55 °C for 30 s, and extension at 72 °C for 45 s, followed by a final extension at 72 °C for 10 min and holding at 4 °C. All amplifications were performed on a T100 Thermal Cycler (Bio-Rad, Hercules, CA, USA).

PCR products were separated on a 2% agarose gel, purified using AMPure^®^ PB beads (Pacific Biosciences, Menlo Park, CA, USA), and quantified with a Qubit 4.0 fluorometer (Thermo Fisher Scientific, USA). Equimolar amounts of purified amplicons were pooled and subjected to paired-end sequencing on the Illumina NextSeq 2000 platform (Illumina, San Diego, CA, USA) by Majorbio Bio-Pharm Technology Co., Ltd. (Shanghai, China), following standard protocols. The raw sequencing data have been submitted to the NCBI Sequence Read Archive (SRA) under Bioproject PRJNA1241221, with accession numbers SRR32839378–SRR32839409.

### 4.4. Bioinformatic Analysis

Raw FASTQ files were demultiplexed using a custom Perl script. Quality control, including adapter trimming and merging of paired-end reads, was performed with fastp (v0.19.6) and FLASH (v1.2.7) [63]. High-quality merged sequences were dereplicated, and chimeras were removed prior to clustering. Operational taxonomic units (OTUs) were clustered at 97% similarity using the UPARSE pipeline (v7.1), with the most abundant sequence selected as the representative for each OTU. Chloroplast-derived sequences were filtered out. Taxonomic assignment of bacterial 16S rRNA gene sequences was performed using the RDP Classifier (v11.5) with a confidence threshold of 0.7. Fungal ITS sequences were annotated by performing a BLASTn search (using the NCBI BLAST+ command line tools, v2.13.0) ) against the UNITE database (v8.0). Alpha diversity indices (Chao1 and Shannon) were calculated based on a rarefied OTU table.

### 4.5. Statistical Analysis

Statistical analyses were performed on the Majorbio Cloud Platform (https://cloud.majorbio.com, accessed on 1 October 2025). Differences in alpha diversity between groups were assessed using the Kruskal–Wallis test, followed by Dunn’s post hoc test with Bonferroni correction [64]. Beta diversity was analyzed via principal coordinate analysis (PCoA) based on Bray–Curtis and Unifrac distances, with the significance of group differences tested using permutational multivariate analysis of variance (PERMANOVA) in the R Vegan package (v2.5−3).

To identify differentially abundant taxa, low-abundance OTUs (<1% of total reads) were filtered out using the summarize_taxa.py script in QIIME (v1.9.1). Group significance was tested with the group_significance.py using Kruskal–Wallis tests with Bonferroni correction (1000 permutations). The core bacterial microbiota was characterized by classifying OTUs into three categories according to their detection frequency across samples: transient (detected in ≤20% of samples), intermediate (>20% and <80%), and persistent (≥80%). The number of OTUs and their cumulative relative abundance were summarized for each category. Community differences among these categories were tested by PERMANOVA. Putative bacterial ecological functions were predicted using the FAPROTAX database (v1.2.1). Microbial composition and differential taxa were visualized via Sankey diagrams and abundance bubble plots [65], respectively.

For soil physicochemical properties, differences between habitats were evaluated using one-way ANOVA, followed by Tukey’s HSD post hoc test where applicable, with results visualized in OriginPro 2022 (v2022b, OriginLab Corporation, Northampton, MA, USA).

## 5. Conclusions

This study investigated the structure and assembly mechanisms of bacterial and fungal communities in the rhizosheath of the desert grass *L. racemosus* under environmentally homogeneous conditions. Our results demonstrate that bacterial and fungal communities exhibit distinct responses to geographic distance and microenvironmental filtering. In sites with highly similar soil and climatic conditions separated by approximately 17 km, bacterial communities showed no significant geographic differentiation. Instead, their composition was predominantly shaped by the rhizosheath microenvironment, which independently explained 46.6% of the variance. In contrast, although fungal communities exhibited statistically significant compositional differences between sites, geographic distance accounted for only 9.51% of the variation, while the rhizosheath microenvironment explained 14.2%. These findings reveal fundamental differences in community assembly rules: bacterial communities are strongly filtered by host-related microenvironmental factors, whereas fungal assembly involves greater stochasticity and dependency on unmeasured environmental variables, historical contingencies, or dispersal limitations.

Based on these contrasting patterns, we propose that microorganisms follow a “geographic sensitivity gradient”: bacterial communities display stronger phylogenetic or functional conservatism and are less influenced by geographic isolation, whereas fungal communities are more sensitive to spatial segregation. This conceptual framework underscores the importance of considering taxon-specific traits—such as dispersal capacity, colonization strategy, and niche breadth—in predicting microbial distribution patterns. In summary, within desert plant–microbe systems, the host plant exerts a dominant role in selectively assembling bacterial communities via the rhizosheath microenvironment, while fungal communities are governed by more complex and spatially heterogeneous processes.

## Figures and Tables

**Figure 1 plants-14-03747-f001:**
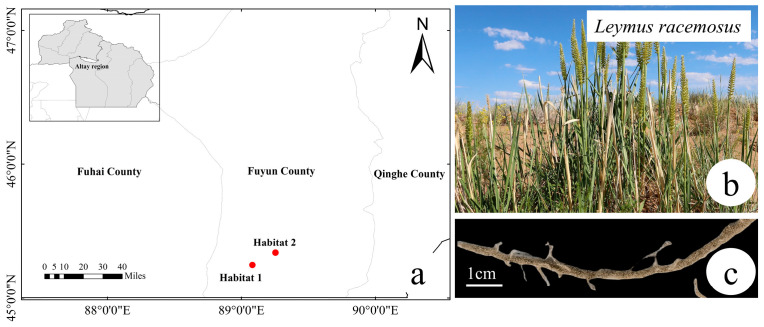
Two primary distribution habitats, Habitat 1 and Habitat 2, of *L. racemosus* in Kalamali Nature Reserve (**a**); images of *L. racemosus* (**b**) and its rhizosheath (**c**).

**Figure 2 plants-14-03747-f002:**
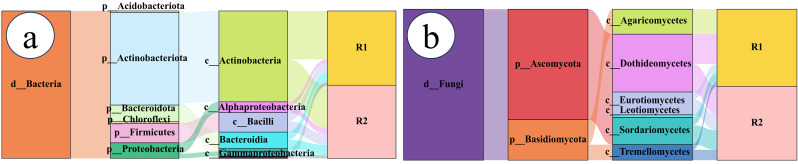
Sankey plots: Flow of the top 50 most abundant bacterial (**a**) and fungal (**b**) taxa at the class level across rhizosheath samples from different habitats. Different colors represent distinct species, and branch width corresponds to the relative abundance of each class; R1 and R2 represent rhizosheath samples collected from Habitat 1 and Habitat 2, respectively.

**Figure 3 plants-14-03747-f003:**
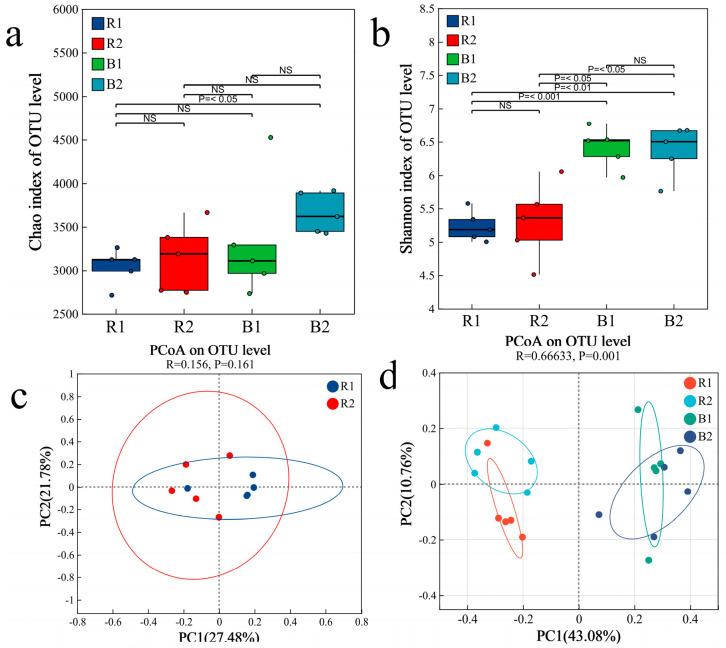
The values of the α−diversity indices for bacterial communities (**a**,**b**) are shown as boxplots for the two habitats. R1 and R2 represent rhizosheath samples collected from Habitat 1 and Habitat 2, respectively; B1 and B2 refer to bulk soil samples obtained from the corresponding habitats; *p* ≤ 0.05, significant; NS, not significant. PCoA showing the bacterial community structure based on distinct samples of the two habitats (**c**) and on rhizosheath and bulk (**d**).

**Figure 4 plants-14-03747-f004:**
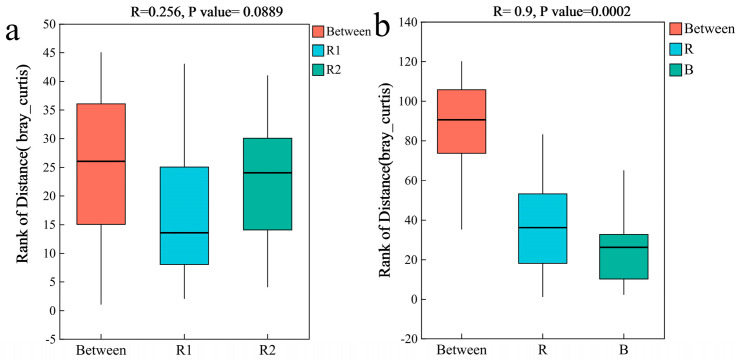
PERMANOVA among different groups. (**a**) Differences in rhizosheath bacteria between the two habitats; (**b**) Differences between rhizosheath and bulk soil bacteria. The R−value is closer to 1, indicating that the difference between groups is more significant than the difference within groups; the smaller R-value indicates no significant difference between and within groups. R1 and R2 represent rhizosheath samples collected from Habitat 1 and Habitat 2, respectively; R: rhizosheath soil samples; B: bulk soil samples.

**Figure 5 plants-14-03747-f005:**
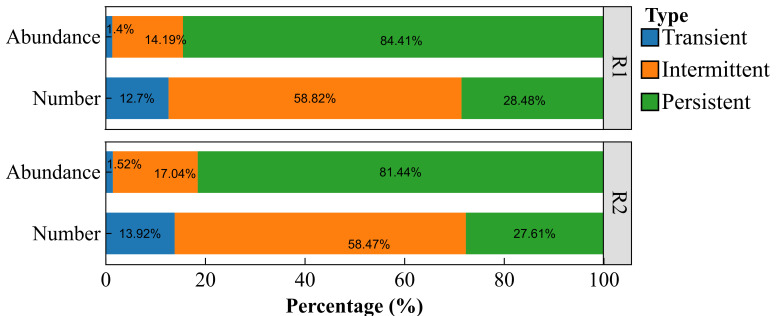
The average relative abundance (Abundance) and detection frequency (Number) of transient, intermediate, and persistent bacterial types in the rhizosheath from Habitat 1 and Habitat 2 (R1, R2). The *x*-axis represents percentage values.

**Figure 6 plants-14-03747-f006:**
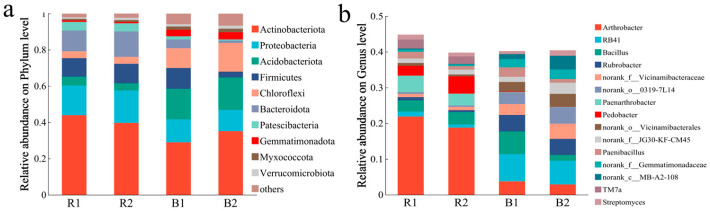
Composition of predominant bacterial phyla in the rhizosheath and bulk soil (**a**) in both habitats, along with the dominant genus within the bacteria (**b**). R1 and R2 represent rhizosheath samples collected from Habitat 1 and Habitat 2, respectively; B1 and B2 refer to bulk soil samples obtained from the corresponding habitats.

**Figure 7 plants-14-03747-f007:**
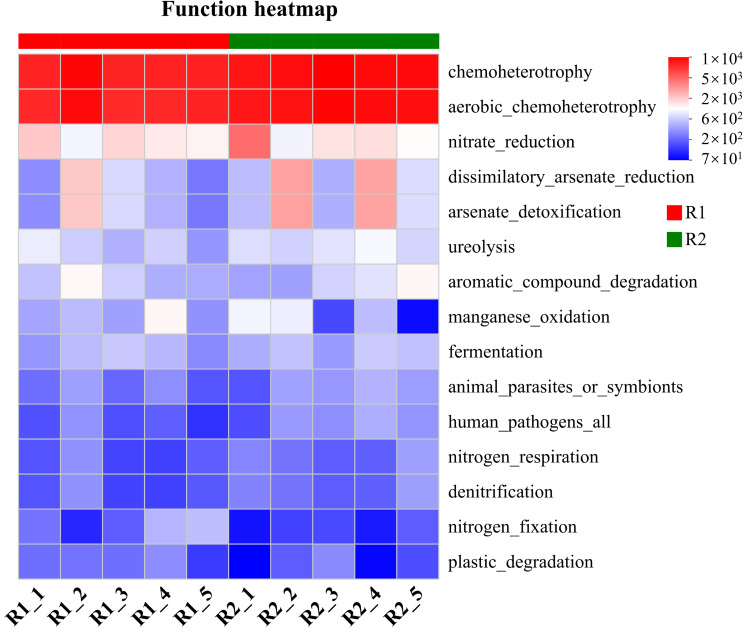
Functional prediction of the rhizosheath microbiota in *L. racemosus* using FAPROTAX. The heatmap displays the abundance profiles of predicted functional groups across different samples. Sample names or group identifiers are shown on the *x*-axis, and the functional traits are listed on the *y*-axis. The color gradient in the tiles represents the relative abundance of each function.

**Figure 11 plants-14-03747-f011:**
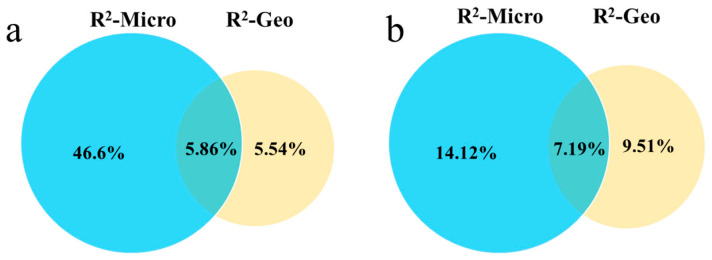
Explanatory power of the *L. racemosus* rhizosheath microenvironment (Micro) and geographical distance (Geo) on bacterial (**a**) and fungal (**b**) communities in the rhizosheath.

**Table 1 plants-14-03747-t001:** Abiotic characteristics of bulk soil in two habitats.

Sample Name	B1	B2	Sample Name	B1	B2
pH	8.17 ± 0.25 ^a^	8.23 ± 0.23 ^a^	W_K_ (mg/kg)	4.18 ± 1.77 ^a^	6.54 ± 1.23 ^a^
EC	47.37 ± 4.32 ^a^	50.17 ± 6.43 ^a^	WNa (mg/kg)	23.66 ± 8.72 ^a^	17.93 ± 3.05 ^a^
TOC (g/kg)	1.03 ± 0.22 ^a^	0.78 ± 0.106 ^a^	WCa (mg/kg)	30.01 ± 5.15 ^a^	25.59 ± 6.05 ^a^
TN (g/kg)	0.06 ± 0.006 ^a^	0.05 ± 0.006 ^a^	SO_4_^2−^ (mg/kg)	15.45 ± 4.65 ^a^	21.32 ± 4.73 ^a^
TP (g/kg)	0.038 ± 0.007 ^a^	0.037 ± 0.005 ^a^	Cl^−^ (mg/kg)	12.92 ± 1.45 ^a^	9.84 ± 2.57 ^a^

B1 and B2 represent the bulk soil from Habitat 1 and Habitat 2, respectively. Letter following the numeral denotes the level of significance of the difference.

**Table 2 plants-14-03747-t002:** PERMANOVA results.

Bacterial				Fungal		
Category	Pseudo-F	R^2^	*p*	Pseudo-F	R^2^	*p*
Geo	0.822	5.54%	0.518	1.472	9.51%	0.079
Micro	12.217	46.6%	0.001	2.321	14.2%	0.004
Geo × Micro	5.534	58.0%	0.001	1.787	30.9%	0.005

## Data Availability

The datasets generated during the current study are available in the NCBI repository, under the Bioproject number PRJNA1241221, available through the web link https://dataview.ncbi.nlm.nih.gov/object/PRJNA1241221 (accessed on 23 March 2025). The corresponding accession numbers for this submission are SRR 32839378–SRR 32839409.

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
