# Peer review of "Niche-Driven Bacterial Assembly Versus Weak Geographical Divergence of Fungi in the Rhizosheath of Desert Plant Leymus racemosus (Lam.) Tzvel"

_plants, 2025, doi:10.3390/plants14243747_

Round 1
Reviewer 1 Report
Comments and Suggestions for Authors
Comments for the author:
This study investigates whether the rhizosheath bacterial and fungal community composition differs between the grass Leymus racemosus originating from two sites. Also, whether the microbial community differs from bulk soil to the rhizosheath. I find this study rather a descriptive study looking at the microbial compositional differences of a single species instead of testing the functions of rhizosheath in plant-microbe interactions. The introduction is very short while missing many important information (i.e. the current knowledge in microbial assembly, the importance of studying the grass Leymus racemosus). Importantly, I think it is not fair to question how geographical distance influence the structure of the rhizosheath bacterial community but only carry on research on two sites. In turn, it is the comparison between two sites. The figures sometimes presented results lacking statistics. Some analyses are not well introduced and failed to provide useful information (i.e., Fig.2 presenting 50 bacterial taxa do not make sense given that the rich taxa of bacteria). There are too many figures showing the same analysis twice for bacteria and fungi separately. Seems that figure 3 and figure 7 have all the information that to answer the question already, the authors should introduce why they need to have the rest analyses. The authors also directly compare microbial composition between bulk soil and rhizosheath, but these samples should have different units for DNA sequencing, is it correct to do this?
Some specific comments:
Introduction: The introduction is a bit too short. The first paragraph outlines the important functions of rhizosheath in belowground recourse uptake, mechanical protection and microbiome selection. But ending the first paragraph with “underscoring its key function in mediating plant-microbe interactions” is not proper, this study did not look at how rhizosheath mediate plant-microbe interactions. Although the second paragraph provides two theories, as the authors already said the important of selection by rhizosheath, why the relative contribution of geographical distance is important? And why we need to know the relative contribution of geographical distance and selection by rhizosheath? The difference between bacteria and fungi seems very important part of the current study, but now this was just briefly mentioned in one sentence. The last paragraph introduces the focal species, which is a narrowly distributed grass, why it is important to use this single species? More information of the focal species such as it is a common or key stone species etc. should be provided. For the question, the authors first ask about the how geographical distance influence the structure of the rhizosheath bacterial community? But there was only one distance, instead of a gradient of distance. I think it can only answer whether the bacterial communities are different between two sites.
Line 42: Better to say, “The role of the rhizosheath as a selective microbial habitat is particularly important”.
Line 51: What do you mean “conserved microbiome composition”?
Line 65: “wo” should be “two”.
Line 73: Overstated. To be honest, I don’t see how a descriptive study can enhance the understanding of plant-microbe interactions.
Fig.1: If you only have two sites in one county, you don’t have to show the geographic figure. Furthermore, (c) more information needs to be provided, i.e. scale under microscope and how did it was sampled as it is only one part of the roots.
Line 83: as here no significant difference, p should above 0.05.
Fig.2: The authors presented the top 50 bacterial and fungal species, but bacteria often have very much more taxa than fungal species. So, is it comparable to set the 50 as cut line? And where shall I see the statistical test?
Line 94: I think if you only check the bacteria at class level, bacterial community of many other grasses will mainly have by Actinobacteria, Alphaproteobacteria, Bacilli, and Bacteroidia.
Line 109: Would that possible to compare bacterial richness between rhizosheath and bulk soil? Given that they probably sampled in different units.
Fig.3: Ideally, panel d should also show the habitat information as well. And you can check if the interaction between the sites and sampling positions (bulk soil and rhizosheath) is significant.
Fig.4: This figure is a bit difficult to follow. What does “Between” mean? Also, different style compared to figure3, as “a” and “b” now are inside the figure.
Fig.5: No idea what “Transient”, “Intermitten” and “Persistent” mean.
Fig.7&Fig.8: same comments as fig.3&4.
Comments on the Quality of English LanguageQuality of English should be improved. There are spelling errors.
Author Response
Please see/find the attached file.

Reviewer 2 Report
Comments and Suggestions for Authors
General
- Too many grammatical mistakes; improve overall writing. Pay attention to spaces between words and between numbers/units.
- Check all affiliations and the corresponding author details. The email address appears incorrect – verify.
Abstract
- Results section is not clearly understandable. PERMANOVA p is non-significant, but the text says “highly conserved.”
Introduction
- Line 65 — “where wo populations?” → clarify and correct to “two populations” (or “two habitats,” if that’s what you mean).
Methods
- Mention how much soil was used for DNA extraction (e.g., 0.25 g per sample) and whether replicates were composited.
- Fungal primer details are missing. Provide primer names, sequences, target region (ITS1 or ITS2).
- Even if using universal primers, add a few lines on PCR conditions (volumes, annealing temperature, cycle number) and whether PCR replicates were performed/pooled.
- If possible, create two subheadings: Bioinformatics and Statistical methods.
- In the bioinformatics section, clearly separate bacteria and fungi. Specify the taxonomy database for each (e.g., SILVA for 16S; UNITE for ITS). It’s currently confusing which steps apply to bacteria vs fungi.
Results
- If possible, rearrange as: 1 Soil properties, 2.2 Microbial community overview (sequence summary + taxa overview) then subsequent subsections.
- Table 1: Footnote not properly stated. Keep a single scheme (letters or asterisks), define it clearly, and remove “(p < 0.05)” from the title if reporting non-significance.
- Figure 2: “species at the class level” — choose one term. If plotted by class, write “classes.”
- Figure 3: Pairwise comparisons done for Shannon but not for Chao1. Keep consistency—either test both or state that only Shannon was tested. Also fix the figure legend formatting.
- Figure 5: “The average relative abundance (Abun) and detection frequency (Num).” Avoid unexplained abbreviations. After this now where its used.
- Section 2.3: Keep p formatting consistent (italic p) throughout; avoid mixing italic and non-italic.
- Figure 5: Mentiom the statistical method used.
- Figure 6: Bacterial panels show Phylum and Genus, but fungal panels use a different style. Keep the visualization consistent across bacteria and fungi .
Author Response
Please see/find the attached file.

Reviewer 3 Report
Comments and Suggestions for Authors
The article presents timely and potentially valuable field research on the rhizosphere microbiome of the desert plant Leymus racemosus. However, several critical components require substantial revision and elaboration to meet the standards of a publishable scientific manuscript. Below are the major concerns:
1. Asymmetry Between Bacterial and Fungal Analyses
While the manuscript is structured to imply a parallel treatment of bacteria and fungi, only the bacterial component is accompanied by a complete methodological description, bioinformatic pipeline, and functional interpretation. The fungal analysis is notably underdeveloped and lacks essential methodological details, including:
ITS primer sequences,
PCR amplification conditions,
sequence quality control and filtering steps,
the taxonomic classification database used (e.g., UNITE).
Given the prominence of fungal community analysis in the results and conclusions, these omissions are critical and must be addressed.
2. Incomplete “Materials and Methods” Section
The ITS sequencing workflow is not described at all.
It is unclear whether standard processing steps (e.g., rarefaction, OTU clustering, normalization) were applied to the fungal dataset as they were for the bacterial 16S data.
While the text discusses functional characteristics of PGPR bacteria, no predictive functional analysis (e.g., PICRUSt, FUNGuild, FAPROTAX) is presented or cited.
3. Limited Statistical Rigor
Statistical tests for differences in taxon abundances (e.g., ANOVA, Kruskal–Wallis, LEfSe) are absent.
The number of biological replicates (n = 3 quadrats per site) is the minimal threshold for ecological inference and should be explicitly acknowledged as a limitation in the Discussion.
Phrases such as "significant difference" appear without accompanying test statistics, p-values, or clarification of the statistical tests used.
4. Insufficient Interpretation and Missing Limitations
The discussion does not clearly distinguish between empirical findings and interpretations based on literature, which may lead to misattribution of evidence.
Alternative explanations—such as stochastic community assembly, priority effects, or seasonal influences—are not considered.
Critical limitations of the study (e.g., lack of functional genomic data, limited sample size, incomplete fungal methodology) are not discussed and should be acknowledged for transparency.
5. Overly General “Discussion” Section
The ecological relevance of dominant fungal taxa (e.g., Talaromyces, Chaetomium) is not sufficiently explored.
Although potential plant-growth-promoting functions are mentioned, no concrete discussion connects these findings to practical applications (e.g., desert ecosystem restoration, sustainable agriculture).
Including a brief paragraph on the applied implications of the findings would enhance the broader impact of the study.
6. Bibliographic Gaps and Inconsistencies
Foundational references on fungal ecology, ITS sequencing, microbial succession, and spatial ecology are lacking.
7. Absence of Explicit Hypotheses
While the manuscript includes general research questions, no formal hypotheses or expected outcomes are stated. Clearly articulated hypotheses would strengthen the conceptual framework of the study.
Summary and Recommendation
This manuscript has clear potential to advance our understanding of microbiome assembly in arid ecosystems. However, major revisions are necessary to ensure methodological transparency, statistical rigor, and conceptual clarity. The following issues should be addressed:
Complete and clarify the methodology for fungal (ITS) sequencing and analysis.
Conduct and report appropriate statistical tests for taxonomic differences.
Explicitly separate data-derived conclusions from speculative or literature-based assertions.
Add a subsection discussing the study’s limitations and potential future directions.
Recommendation: Major Revision
With these substantive revisions, the manuscript could make a meaningful contribution to microbial biogeography and niche theory in extreme environments.
Comments on the Quality of English LanguageWhile the manuscript is generally readable, there are multiple instances where sentence structure, word choice, or transitions could be refined for clarity. In particular, some methodological descriptions are fragmented or overly dense, and parts of the discussion would benefit from more precise academic phrasing. A careful language revision would improve the accessibility and professionalism of the text.
Author Response
Please see/find the attached file.

Round 2
Reviewer 2 Report
Comments and Suggestions for Authors
Dear Editor,
The authors have revised the paper substantially; however, multiple inconsistencies in word fonts appear throughout the revised manuscript and should be corrected. In addition, the citations for References 61 and 62 need to be verified for accuracy.
Author Response
Thank you very much for taking the time to review this manuscript. Your comments and suggestions are invaluable in enhancing the quality of our work. Please find our detailed responses below and the corresponding revisions/corrections highlighted in the resubmitted files.
Comments 1: The authors have revised the paper substantially; however, multiple inconsistencies in word fonts appear throughout the revised manuscript and should be corrected. In addition, the citations for References 61 and 62 need to be verified for accuracy.
Response 1: Thank you for pointing this out. We agree with this comment. Therefore, we have carefully checked and unified the word fonts throughout the entire manuscript to ensure consistency. Regarding the citations for primers, we have also verified References 61 and 62 and replaced them with more accurate and appropriate references as suggested. The modifications can be found on Page 17, Lines 638–641.
Comments 2: Methods can be improved.
Response 2: Thank you for pointing this out. We agree with this comment. Therefore, we have thoroughly revised the Methods section to enhance its clarity and logical flow. Specifically, we have reorganized the structure, removed redundant descriptions, and consolidated fragmented information into coherent paragraphs. These modifications can be found in the Materials and Methods section, spanning from Page 13 to Page 14.
Reviewer 3 Report
Comments and Suggestions for Authors
After reviewing the revised version of the manuscript entitled “Niche-driven bacterial assembly versus weak geographical divergence of fungi in the rhizosheath of the desert plant Leymus racemosus (Lam.) Tzvel,” submitted to the journal Plants, I conclude that:
The authors have addressed the reviewers' comments submitted in the first round of reviews in a satisfactory and substantively accurate manner. The changes introduced significantly improved the quality of the work in terms of clarity, methodological transparency, and interpretation of results.
While the manuscript is generally readable, there are multiple instances where sentence structure, word choice, or transitions could be refined for clarity. In particular, some methodological descriptions are fragmented or overly dense, and parts of the discussion would benefit from more precise academic phrasing. A careful language revision would improve the accessibility and professionalism of the text.
Author Response
Thank you very much for taking the time to review this manuscript. Your comments and suggestions are invaluable in enhancing the quality of our work. Please find our detailed responses below and the corresponding revisions/corrections highlighted in the resubmitted files.
Comments 1: While the manuscript is generally readable, there are multiple instances where sentence structure, word choice, or transitions could be refined for clarity. In particular, some methodological descriptions are fragmented or overly dense, and parts of the discussion would benefit from more precise academic phrasing. A careful language revision would improve the accessibility and professionalism of the text.
Response 1: Thank you for pointing this out. We agree with this comment. Therefore, we have conducted a thorough, word-by-word revision of the entire manuscript to enhance its clarity, flow, and academic tone.
Specifically, in the Materials and Methods (Section 4, Pages 12-14), we have completely reorganized the structure to eliminate fragmentation and reduce density. Redundant descriptions were removed, and related procedures were consolidated into coherent, logically flowing paragraphs.
Furthermore, in the Discussion section (Section 3, Pages 9-12), we have refined the sentence structures and replaced imprecise phrasing with more accurate and formal academic terminology. Special attention was paid to strengthening the logical transitions between paragraphs and sharpening the interpretation of our findings.
We believe these comprehensive revisions have significantly improved the overall readability, professionalism, and scholarly impact of the manuscript.